# VIS-SLAM: A Real-Time Dynamic SLAM Algorithm Based on the Fusion of Visual, Inertial, and Semantic Information

Yinglong Wang [1], Xiaoxiong Liu [2,*], Minkun Zhao [1] and Xinlong Xu [1]

1   School of Automation, Northwestern Polytechnical University, Xi'an 710072, China;
    wyl4435@mail.nwpu.edu.cn (Y.W.); minkun_zhao@mail.nwpu.edu.cn (M.Z.); xuxinlong@mail.nwpu.edu.cn (X.X.)
2   Shaanxi Province Key Laboratory of Flight Control and Simulation Technology, Xi'an 710072, China
*   Correspondence: nwpulxx@outlook.com

**Abstract:** A deep learning-based Visual Inertial SLAM technique is proposed in this paper to ensure accurate autonomous localization of mobile robots in environments with dynamic objects. Addressing the limitations of real-time performance in deep learning algorithms and the poor robustness of pure visual geometry algorithms, this paper presents a deep learning-based Visual Inertial SLAM technique. Firstly, a non-blocking model is designed to extract semantic information from images. Then, a motion probability hierarchy model is proposed to obtain prior motion probabilities of feature points. For image frames without semantic information, a motion probability propagation model is designed to determine the prior motion probabilities of feature points. Furthermore, considering that the output of inertial measurements is unaffected by dynamic objects, this paper integrates inertial measurement information to improve the estimation accuracy of feature point motion probabilities. An adaptive threshold-based motion probability estimation method is proposed, and finally, the positioning accuracy is enhanced by eliminating feature points with excessively high motion probabilities. Experimental results demonstrate that the proposed algorithm achieves accurate localization in dynamic environments while maintaining real-time performance.

**Keywords:** SLAM; dynamic objects; deep learning; non-blocking model; inertial; real time

## 1. Introduction

As a foundational research in many emerging technologies, simultaneous localization and mapping (SLAM) techniques have experienced unprecedented development in recent years and have found widespread applications in fields such as autonomous driving, augmented reality, and unmanned aerial vehicles. Considering the advantages of the low cost and high portability of visual cameras, visual SLAM techniques have received significant attention over the past few decades. However, most existing SLAM techniques make a strong assumption of a static environment [1]. In the real world, there are constantly numerous moving objects present, such as pedestrians in shopping malls and on roads, fast-moving vehicles, and other potential moving entities. These objects have adverse effects on pose estimation. Although pure visual methods, such as the epipolar geometry constraint, RANSAC [2], PROSAC [3], and multi-view geometry, can partially suppress the influence of dynamic objects, such algorithms can introduce serious errors when a large number of moving targets or high-motion objects are present in the environment, leading to the system becoming overwhelmed. Therefore, improving the dynamic robustness of SLAM systems is of utmost importance. The current mainstream solution for dynamic environments is to remove dynamic features that affect localization and mapping and only utilize static points in the environment for positioning and mapping. This paper categorizes existing visual localization algorithms in dynamic scenes into three major classes.

The first category of methods is based on pure visual geometry constraints for dynamic SLAM. These algorithms determine the dynamic or static nature of feature points by examining the consistency of geometric motion between image frames, and subsequently filter

out dynamic points. VINS-Mono [4] computes the fundamental matrix using the Random Sample Consensus (RANSAC) algorithm to remove outliers. Sun [5] utilizes RANSAC for reprojection error measurement to reject dynamic points and enhances motion object detection using particle filtering. Liu proposes a dynamic object detection algorithm based on RGB-D cameras, which computes dense optical flow to estimate the pose transformation between image frames and constructs the reprojection error to estimate the dynamic region. Wang [6] presents a dynamic SLAM algorithm based on epipolar geometry constraints and mathematical models, but it suffers from significant localization errors in highly dynamic scenes due to heavy reliance on transformation matrix estimation. Zhang [7] introduces the PFD-SLAM algorithm, which utilizes a non-prior semantic segmentation algorithm based on particle filtering. It computes the discrepancy between grid-based motion statistics and optical flow calculations as the observation for posterior estimation, then employs RANSAC to extract dynamic regions based on the homography matrix and, finally, applies particle filtering. Dai [8] determines the state of feature points based on data association and uses the Delaunay [9] triangulation method to deploy discrete map points into a sparse map. This category of methods offers a fast computation speed and high real-time performance, but cannot eliminate features on potential moving objects.

The second category of methods is based on deep learning for dynamic SLAM techniques. These methods utilize deep learning techniques to obtain per-pixel masks or bounding boxes of potential moving objects, generating mask information to remove dynamic features from these objects and improve localization accuracy. Compared to traditional image processing algorithms and geometry constraint methods, deep learning-based dynamic SLAM methods can leverage higher level feature information to determine the dynamic or potential motion attributes of feature points. Moreover, deep learning is widely applied in backend processing, such as deep learning-based loop-closure-detection algorithms, greatly improving the efficiency and localization accuracy of backend optimization. Yu [10] proposes DS-SLAM, which is a dynamic SLAM algorithm based on the integration of ORB-SLAM2 [11] and the SegNet [12] semantic segmentation network. Bescos [13] introduces DynaSLAM based on the Mask-RCNN network, which not only provides dynamic–static prior information through semantic segmentation, but also utilizes multi-view geometry methods for calibration, demonstrating excellent performance in terms of localization accuracy. However, the complex structure of the Mask-RCNN network leads to poor real-time performance. To improve computational efficiency, Liu [14] presents a real-time dynamic SLAM algorithm called RDS-SLAM, which employs a non-blocking model that performs semantic segmentation only on keyframes, while the remaining image frames are processed using a Bayesian probability model for feature prediction. This approach significantly enhances real-time performance. However, since only keyframes are used to determine the corresponding map points, the real-time output results are not ideal.

The third category of methods is based on the fusion of multiple sensor modalities for dynamic SLAM algorithms. Visual cameras, as active sensors, are susceptible to changes in the environment, which can affect the quality of the input and, consequently, impact localization accuracy. However, there are sensors like the Inertial Measurement Unit (IMU), whose inputs are less affected by environmental changes. By fusing such sensors with visual cameras, the interference of environmental changes on the localization system can be mitigated to some extent. Zhao [15] deploys the DeepLabv3 [16] network into the VINS-Mono system, combining semantic segmentation and motion-consistency detection algorithms to remove dynamic features. Song [17] proposes a dynamic SLAM algorithm based on visual–inertial sensors, called DynaVINS. This algorithm does not employ deep learning theory, but utilizes visual–inertial fusion to make joint judgments on feature information, assigning different weights to features based on their dynamic nature, with stronger dynamics having smaller weights. Liu [18] introduces a dynamic SLAM algorithm based on RGB-D Inertial, which uses YOLOv5 [19] as the object-detection algorithm and combines IMU motion consistency detection and epipolar geometry constraint methods to reject

dynamic features. Sun [20] presents the D-VINS algorithm, which combines deep learning, IMU consistency detection, and geometric constraints to estimate the probability of feature point motion and assigns different weights to feature points based on this probability. Although these algorithms improve robustness by leveraging the advantages of sensor fusion, determining reasonable ranges for some hyperparameters in the algorithms, such as the segmentation threshold in IMU consistency detection, can be challenging.

In response to the above-mentioned issues, this paper proposes real-time visual–inertial odometry for resource-constrained aerial vehicles in dynamic environments. In this system, semantic information is first extracted from image frames based on a non-blocking model. Then, motion segmentation is performed using a motion probability grading model and motion probability propagation model. Inertial measurement information is introduced to correct errors introduced by the coarse segmentation algorithm, thereby improving localization accuracy in dynamic environments. The main contributions of this paper are as follows:

(1) We propose a method for extracting semantic information in a non-blocking manner, thereby reducing the impact of the deep learning module on real-time system performance and improving the algorithm's execution speed. Furthermore, we design a mobile probability grading model and a mobile probability propagation model for performing coarse motion segmentation.

(2) A visual–inertial tightly coupled feature optimization algorithm is proposed, leveraging prior semantic information. An adaptive threshold-based motion consistency detection algorithm is designed to refine the coarse segmentation results.

(3) We conducted experiments on datasets and in real environments. The results demonstrate that the proposed algorithm achieves good localization accuracy in highly dynamic environments.

## 2. System Overview

With the aim of improving the positioning accuracy and algorithmic robustness of visual localization algorithms in dynamic environments, this paper proposes two key modules. The first module is the motion probability coarse estimation module, which utilizes the YOLOv8 network [21] to extract semantic information from images and generate corresponding semantic masks. Different motion probabilities are assigned to feature points located in different mask regions.

The module adopts a non-blocking architecture to enhance real-time performance, meaning that semantic information extraction operations are not performed for every frame of the image. Instead, when the semantic thread is idle, the system detects the latest frame in the sliding window and extracts semantic information only for that frame. For frames without extracted semantic information, feature point motion probabilities can be estimated using epipolar geometry constraint algorithms and motion probability propagation models. Considering the weak robustness of pure visual SLAM algorithms and the unaffected output of IMU sensors by dynamic objects, this paper further investigates a method that combines visual and IMU information to determine the static or dynamic nature of feature points. This method is implemented in the second module, the feature optimization algorithm based on visual–inertial coupling. The module utilizes the prior information provided by the coarse segmentation module to perform motion consistency checks in order to correct the erroneous motion probability estimates of feature points from the coarse segmentation module. This correction process improves the positioning accuracy of the algorithm. The system framework diagram is shown in Figure 1.

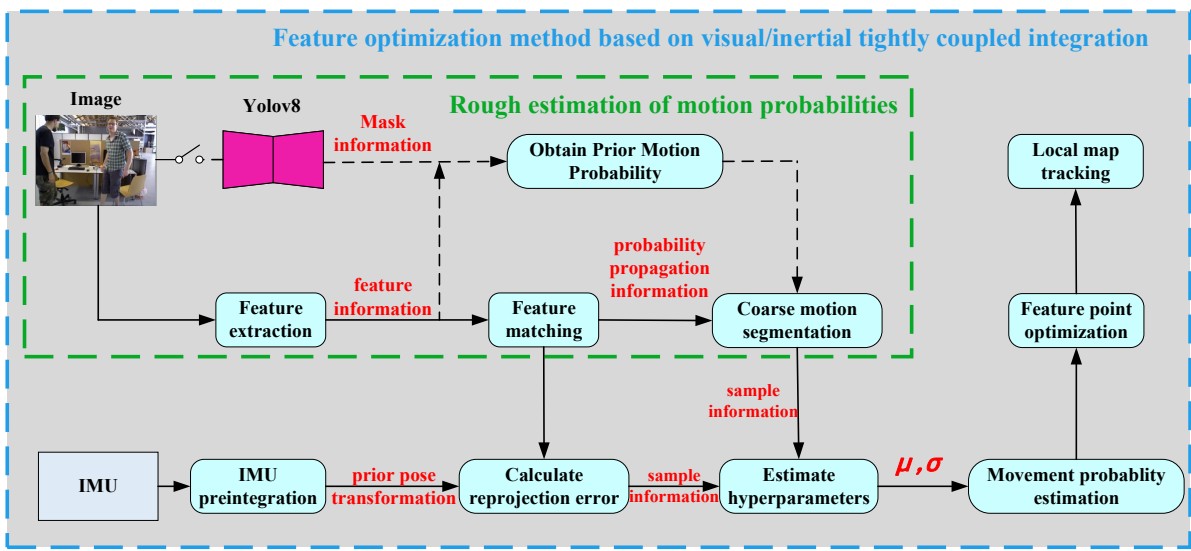

**Figure 1.** System framework diagram.

## 3. Methods

### 3.1. Deep Learning-Based Motion Coarse Estimation Algorithm

The deep learning-based image segmentation method is a commonly used approach to improve the accuracy of SLAM localization in dynamic environments. However, most of these methods are blocking models, meaning that, after extracting feature points from each frame, the system needs to wait for the results of semantic extraction in order to optimize the features based on the semantic information. This dependence on the speed of the deep learning module significantly affects the system's overall speed, making high demands on computational resources. Typically, the speed of semantic information extraction is much slower than that of feature point extraction. This forces the feature-extraction thread to wait for the results from the semantic thread, greatly reducing the real-time performance of the localization system. While RDS-SLAM proposes a non-blocking model that only performs semantic extraction on keyframes, this approach operates on map points, which may be discarded, and its method of segmenting keyframes and removing dynamic feature points does not dynamically improve feature matching accuracy in real time, making it a lagging optimization method.

#### 3.1.1. Non-Blocking Semantic Information-Extraction Algorithm Design

To improve the efficiency of feature point optimization, this paper adopts a parallel semantic information-extraction thread parallel to the feature-extraction thread, drawing inspiration from conventional semantic SLAM models [22]. Utilizing the concept of a sliding window model [4], when extracting features from all frames within the sliding window, the semantic information of only the most recent frame in the window needs to be extracted. As shown in Figure 2, assuming the time for feature extraction per frame is $\Delta t$ and the time for semantic information extraction per frame is $\Delta T$, if a blocking model is used, after the feature extraction of frame $n$ is completed, the waiting time required would be $n(\Delta T - \Delta t)$. As time progresses, the accumulation of the waiting time becomes significant, leading to a loss of real-time tracking capability. In contrast, when a non-blocking model is used, only the semantic information of the latest frame in the sliding window needs to be extracted. Consequently, the waiting time becomes a fixed value, $\Delta T_g = \Delta T - \Delta t$, and does not accumulate over time. Frames that have not undergone semantic extraction will be classified based on the semantic information extracted from the frames within the sliding window. Compared to the blocking model, the non-blocking model imposes lower performance requirements on the onboard processor. Even if the

processor fails to perform semantic segmentation, thus affecting the tracking process, it does not impact the overall system performance.

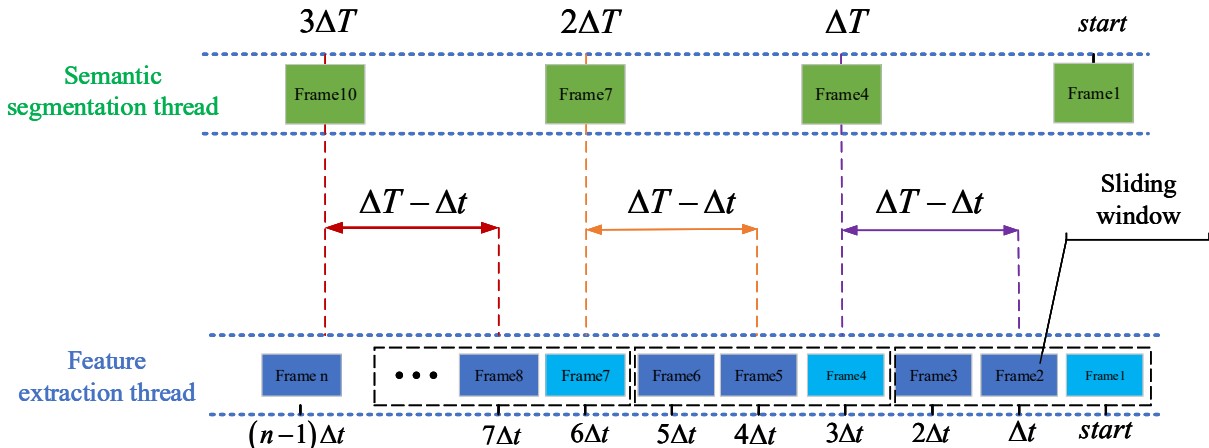

**Figure 2.** Non-blocking semantic information extraction diagram.

Through object detection algorithms, semantic information can be obtained from images. Based on this semantic information, initial determinations about the static or dynamic nature of feature points can be made. For instance, when dynamic objects such as pedestrians or fast-moving trains are detected in an image, the relationship between the pixel positions of feature points and the bounding boxes of these objects can be calculated to determine their static or dynamic nature. If a feature point falls within a bounding box corresponding to a dynamic object, it is considered a dynamic point; otherwise, it is considered a static feature.

### 3.1.2. Motion Probability Grading and Propagation Model

While deep learning algorithms can provide semantic information from images, directly using this information to determine the static or dynamic nature of objects may not be entirely reliable. For example, books and tables in a classroom are generally stationary, but when someone moves them, they become dynamic objects. Similarly, in a shopping mall, most customers are in motion, but some may be stationary. Therefore, directly assigning static or dynamic properties to feature points based on semantic information may not be applicable in certain scenarios. To address this issue, this section proposes a motion probability grading model. Drawing from real-world experience, this model assigns different prior motion probabilities to different semantic types of objects, rather than simply classifying them as static or dynamic. This model is then combined with the previously mentioned epipolar geometry constraint algorithm to determine the static or dynamic nature of feature points.

In this paper, based on life experience, we have categorized the labels provided by the COCO dataset [23] into five major classes, where the higher the probability of movement, the higher the level, and conversely, the smaller the probability of movement, the lower the level. As shown in Figure 3, the first category is designated as public facilities, including objects like traffic lights, signs, and benches. The second category includes household or office items such as books, desks, chairs, and sofas.

The third category encompasses transportation vehicles like buses and bicycles. The fourth category includes humans and sports equipment. The final category consists of animals like birds, cats, and dogs. For feature points in the images, their prior motion probabilities are assigned based on the different semantic masks they fall into. If a feature point falls into an area without a semantic mask, that area is considered a background region, and such feature points are directly classified as static features, which can be used directly for pose estimation.

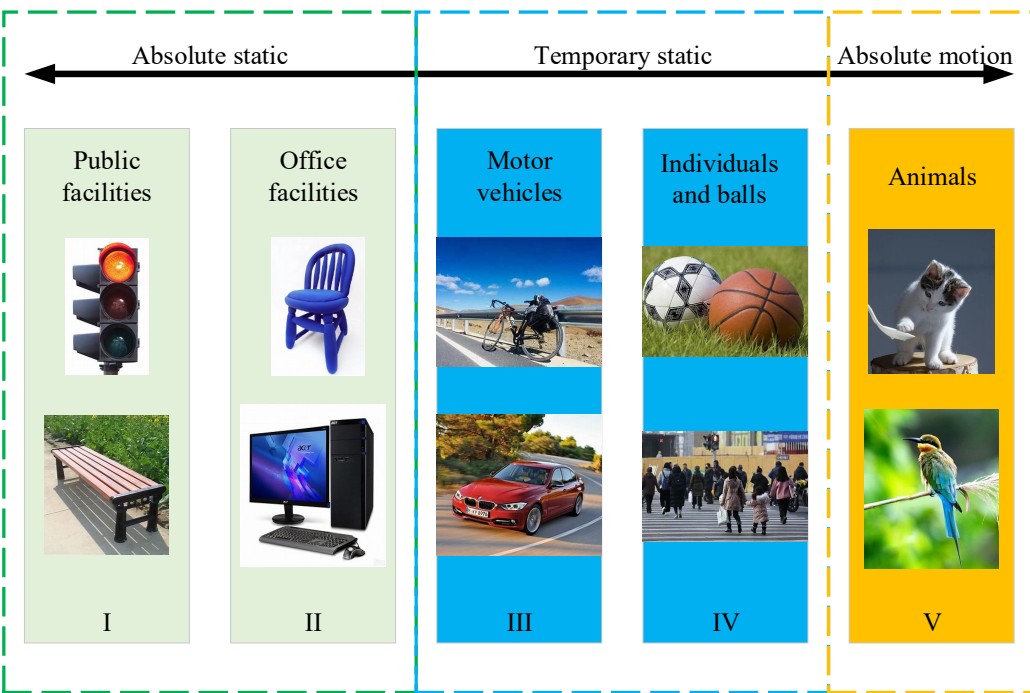

**Figure 3.** Motion probability grading model.

Generally speaking, if the detected semantic objects are traffic lights or buildings, the feature points located on them can be considered static feature points and used for visual pose estimation and map construction. On the other hand, if the detected objects are birds or livestock, considering that they are often in motion and occupy a small area in the image, their impact on localization or mapping is relatively small. Therefore, feature points located on them can be classified as dynamic features. However, under normal circumstances, the semantic extraction network is unable to determine the semantic attributes of each feature point. For these feature points, we follow the idea of the ORB-SLAM3 algorithm and categorize them as background features, that is static features.

### 3.1.3. Motion Probability Propagation Model

In the non-blocking semantic information extraction model, there are cases where some image frames do not have extracted semantic information, resulting in feature points on these frames lacking prior motion probability. To ensure that each image frame has prior semantic information, this section proposes a motion probability propagation model. For image frames without extracted semantic information, their own image feature's motion probability is estimated based on the feature matching relationships with other frames in the sliding window and the semantic information from those frames. The motion probability for the five categories is defined as Equation (1):

$$
s_{p_i}^t = \begin{cases} 1, p_i \in \Omega(\text{V}) \\ 0.75, p_i \in \Omega(\text{IV}) \\ 0.5, p_i \in \Omega(\text{III}) \\ 0.25, p_i \in \Omega(\text{II}) \\ 0, p_i \in \Omega(\text{I}) \end{cases} \tag{1}
$$

In the equation, $s_{p_i}^t$ represents the motion probability of feature points at time $t$; different semantic categories have different motion probabilities, with higher categories having higher probabilities and lower categories having lower probabilities. $\Omega$ represents the feature space, so the motion probability of the feature point set at time $t$ can be defined as $S_t = \{s_{p_1}^t, s_{p_2}^t, \ldots\ldots, s_{p_n}^t\}$.

In the non-blocking semantic information extraction model, due to the higher speed of feature extraction compared to semantic information extraction, there will inevitably be some image frames without extracted semantic information. For these frames, this paper proposes using their adjacent frames and Bayesian estimation methods [24] to determine the prior motion probability of feature points in the image, The specific calculation method is shown in Equation (2):

$$bel(s_{p_i}^t) = p(s_{p_i}^t|Z_{1:t}, s_{p_i}^0) \propto p(Z_t|s_{p_i}^t, Z_{1:t-1}, s_{p_i}^0)p(s_{p_i}^t|Z_{1:t-1}, s_{p_i}^0) \tag{2}$$

In the equation, $s_{p_i}^0$ represents the initial motion probability of the feature points, which is obtained through Equation (1) and $Z$ represents inter-frame matching observations. At this point, the entire state estimation can be transformed into a maximum a posteriori probability estimation, as shown in Equation (3):

$$\begin{aligned} \arg\max p(s_{p_i}^t|Z_{1:t}, s_{p_i}^0) &= \arg\max p(Z_t|s_{p_i}^t, Z_{1:t-1}, s_{p_i}^0)p(s_{p_i}^t|Z_{1:t-1}, s_{p_i}^0) \\ &= \arg\max p(Z_t|s_{p_i}^t)p(s_{p_i}^t|Z_{1:t-1}, s_{p_i}^0) \\ &= \arg\max p(Z_t|s_{p_i}^t) \int p(s_{p_i}^t|s_{p_i}^{t-1}, Z_{1:t-1})p(s_{p_i}^t|Z_{1:t-1})ds_{p_i}^{t-1} \end{aligned} \tag{3}$$

By solving the above maximum a posteriori estimation problem, the prior motion probability of feature points in image frames without extracted semantic information can be calculated.

### 3.2. Visual–Inertial Tight-Coupling Feature Optimization Algorithm

Visual information serves as the foundation of VSLAM, providing rich data that, in the context of a local world coordinate system, allows for real-time output of the camera's absolute pose. However, visual data have a lower sampling frequency and are susceptible to factors such as field of view and environmental conditions. In contrast, IMU sensors have a higher sampling frequency and are less affected by environmental factors. However, IMU sensors only provide the carrier's acceleration and angular velocity information, requiring continuous integration to output the carrier's absolute pose. Nevertheless, prolonged operation inevitably leads to accumulated errors, resulting in the divergence of the localization result. To address these challenges, VIO algorithms integrate visual and IMU information to compensate for their respective limitations, thereby achieving high-precision localization.

#### 3.2.1. IMU Preintegration Model

The IMU measures a moving vehicle's inertial forces to obtain its own three-axis acceleration and three-axis angular velocity. By continuously integrating these measurements, the IMU can provide the vehicle's velocity, position, and Euler angles. Typically, an IMU consists of two main components: an accelerometer for measuring three-axis acceleration and a gyroscope for measuring three-axis angular velocity. The IMU used in this paper is a low-cost MEMS IMU, known for its relatively lower measurement accuracy and higher noise levels, which can lead to error accumulation and drift.

Due to the relatively low speeds of the motion carriers in this paper, effects such as centripetal acceleration and Coriolis acceleration caused by the Earth's rotation can be neglected. Therefore, the output of the IMU is mainly affected by measurement noise and bias. Assuming that the measurement noises of the accelerometer and gyroscope are $\eta_{acc}$ and $\eta_{gyro}$, respectively, and their biases are $\boldsymbol{b}_{acc}$ and $\boldsymbol{b}_{gyro}$, the IMU measurement model can be established as Equation (4):

$$\begin{cases} \hat{\boldsymbol{a}}^{b_t} = \boldsymbol{a}^{b_t} - \boldsymbol{R}_w^{b_t}\boldsymbol{g}^w + \boldsymbol{\eta}_{acc} + \boldsymbol{b}_{acc} \\ \hat{\boldsymbol{\omega}}^{b_t} = \boldsymbol{\omega}^{b_t} + \boldsymbol{b}_{gyro} + \boldsymbol{\eta}_{gyro} \end{cases} \tag{4}$$

In the equation, $\hat{a}^{b_t}$ represent the measured values of the accelerometer and gyroscope at time $t$, respectively, while $a^{b_t}$ and $\omega^{b_t}$ represent the true acceleration and true angular velocity of the IMU at time $t$. The IMU's bias can be considered a Wiener process, also known as a random walk or Brownian motion, with its derivative following a Gaussian distribution [25].

According to the above measurement model, the continuous-time form of the IMU motion can be described as Equation (5):

$$
\begin{cases}
\dot{p}_{b_t}^w = v_{b_t}^w \\
\dot{v}_{b_t}^w = a_{b_t}^w \\
\dot{q}_{b_t}^w = \dfrac{1}{2} q_{b_t}^w \otimes [0, \omega_{b_t}]^T
\end{cases}
\tag{5}
$$

In the equation, $p_{b_t}^w$, $v_{b_t}^w$, $a_{b_t}^w$, $q_{b_t}^w$,, $\omega^{b_t}$, respectively, represent the position, velocity, acceleration, quaternion attitude, and angular velocity of the IMU relative to a given world coordinate system at time. Define the time interval $[t_i, t_{i+1}]$. Based on the above equation, the motion equation from time $t_i$ to time $t_{i+1}$ can be determined as Equation (6).

$$
\begin{cases}
p_{b_{i+1}}^w = p_{b_i}^w + v_{b_i}^w \Delta t_i + \iint_{t \in [t_i, t_{i+1}]} [R_{b_t}^w(\hat{a}^{b_t} - b_a^{b_t} - n_a) + g^w] dt^2 \\
v_{b_{i+1}}^w = v_{b_i}^w + \int_{t \in [t_i, t_{i+1}]} [R_{b_t}^w(\hat{a}^{b_t} - b_a^{b_t} - n_a) + g^w] dt \\
q_{b_{i+1}}^w = \int_{t \in [t_i, t_{i+1}]} \dot{q}_{b_t}^w dt
\end{cases}
\tag{6}
$$

In order to avoid repeated integration during each optimization process, the IMU preintegration theory was proposed in reference [26]. Based on Equation (6), the inter-frame IMU pre-integration result can be obtained as Equation (7):

$$
\begin{cases}
\alpha_{b_{i+1}}^{b_i} = \iint_{t \in [t_i, t_{i+1}]} R_{b_t}^{b_i}(\hat{a}^{b_t} - b_a^{b_t} - n_a) dt^2 \\
\beta_{b_{i+1}}^{b_i} = \int_{t \in [t_i, t_{i+1}]} R_{b_t}^{b_i}(\hat{a}^{b_t} - b_a^{b_t} - n_a) dt \\
q_{b_{i+1}}^{b_i} = \int_{t \in [t_i, t_{i+1}]} \dfrac{1}{2} \Omega(\hat{\omega}^{b_t} - b_g^{b_t} - n_g) q_{b_t}^{b_i}
\end{cases}
\tag{7}
$$

Based on the IMU preintegration result, the pose transformation matrix between adjacent frames can be obtained as Equation (8):

$$
T_{b_{i+1}}^{b_i} = \begin{bmatrix} R_{b_{i+1}}^{b_i} & t \\ 0 & 1 \end{bmatrix}
\tag{8}
$$

In the equation, $R_{b_{i+1}}^{b_i}$ can be obtained by transforming the quaternion $q_{b_{i+1}}^{b_i}$ and the translation vector $t = p_{b_{i+1}}^{b_i}$.

### 3.2.2. The Motion-Consistency-Detection Algorithm Based on Visual–Inertial Measurement Unit

Generally speaking, when a feature point is static, its reprojection error should be close to zero due to satisfying the epipolar geometric constraint. However, when the feature point moves, the epipolar geometric constraint is no longer satisfied, and thus, the reprojection error is no longer zero, increasing with the magnitude of the movement. Assuming the coordinates of a 3D point in the local world coordinate system are $p_k^w = [x_k, y_k, z_k]^T$, and this point is observed by both adjacent frames $F_i$ and $F_{i+1}$, the following constraint exists for the landmark point:

$$\begin{cases} \boldsymbol{T}_w^{C_i}\boldsymbol{p}_k^w = \boldsymbol{p}_k^{C_i} \\ \boldsymbol{T}_w^{C_{i+1}}\boldsymbol{p}_k^w = \boldsymbol{p}_k^{C_{i+1}} \end{cases} \tag{9}$$

In the equation, $\boldsymbol{T}_w^{C_i}$ and $\boldsymbol{T}_w^{C_{i+1}}$, respectively, represent the coordinate transformation matrices from the world coordinate system to the image frame $F_i$ and $F_{i+1}$ the corresponding camera coordinate system. $\boldsymbol{p}_k^{C_i}$ and $\boldsymbol{p}_k^{C_{i+1}}$, respectively, represent the 3D coordinates of the landmark point $k$ in the corresponding camera coordinate system to the images $F_i$ and $F_{i+1}$. Based on the above equation, the constraint relationship between them can be obtained as Equation (10):

$$\boldsymbol{p}_k^{C_{i+1}} = \boldsymbol{T}_{b_{i+1}}^{C_{i+1}}\boldsymbol{T}_{b_{i+1}}^{b_i}\boldsymbol{T}_{b_i}^{C_i}\boldsymbol{p}_k^{C_i} \tag{10}$$

$\boldsymbol{T}_{b_{i+1}}^{C_{i+1}}$ and $\boldsymbol{T}_{b_i}^{C_i}$, respectively, represent the camera–IMU coordinate transformation matrix at different times, which is usually considered to remain unchanged with the motion of the carrier. As shown in Figure 4a, for a 3D point in the world coordinate system, due to its displacement, it no longer satisfies the epipolar geometric constraint with the projected points in images $F_i$ and $F_{i+1}$ leading to the appearance of reprojection errors (highlighted in red).

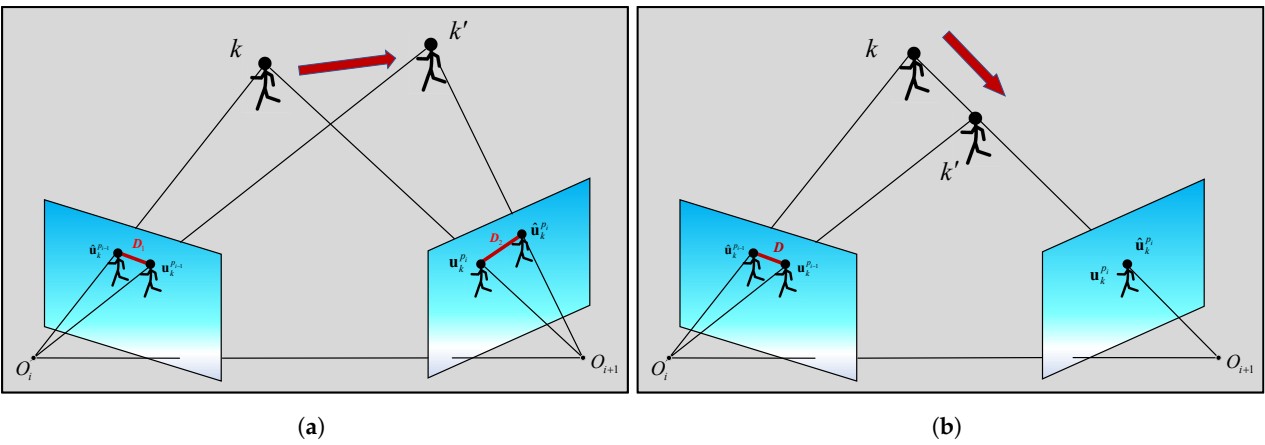

        (**a**)                                            (**b**)

**Figure 4.** The reprojection error model for dynamic features. (**a**) Moving in a non-epipolar direction; (**b**) moving in an epipolar direction.

Constructing the reprojection error of the co-visible feature points in the image frame $F_{i+1}$, we can obtain Equation (11):

$$r_k = \| \hat{\boldsymbol{u}}_k^{p_{i+1}} - \boldsymbol{u}_k^{p_{i+1}} \|_2^2 = \| \hat{\boldsymbol{u}}_k^{p_{i+1}} - \boldsymbol{\pi}(\boldsymbol{T}_b^C\boldsymbol{T}_{b_i}^{b_{i+1}}\boldsymbol{T}_C^b\boldsymbol{p}_k^{C_i}) \|_2^2 \tag{11}$$

In the equation, $\hat{\boldsymbol{u}}_k^{p_{i+1}}$ represents the observed coordinates of a feature point in the pixel coordinate system of the image frame $F_{i+1}$, $\boldsymbol{u}_k^{p_{i+1}}$ represents the theoretical coordinates of the feature point in the pixel coordinate system of the image frame $F_{i+1}$, which is the inter-frame projection result, and $\boldsymbol{\pi}(*)$ represents the forward projection function from the camera coordinate system to the pixel coordinate system. However, in certain motion scenarios, using the reprojection error of feature points to determine their static or dynamic nature may not be effective. As shown in Figure 4b, when a dynamic object moves toward or away from the optical center of the camera, the reprojection error remains zero regardless of the direction of motion. To address this issue, this paper proposes a bidirectional projection method: for the same feature point, it is necessary not only to compute its reprojection error in the current frame, but also to obtain its reprojection error in the previous frame through IMU preintegration. The calculation formula in this case is as Equation (12):

$$\begin{aligned} r_k &= \| \hat{\boldsymbol{u}}_k^{p_i} - \boldsymbol{u}_k^{p_i} \|_2^2 + \| \hat{\boldsymbol{u}}_k^{p_{i-1}} - \boldsymbol{u}_k^{p_{i-1}} \|_2^2 \\ &= \| \hat{\boldsymbol{u}}_k^{p_i} - \boldsymbol{\pi}(\boldsymbol{T}_b^C\boldsymbol{T}_{b_{i-1}}^{b_i}\boldsymbol{T}_C^b\boldsymbol{p}_k^{C_{i-1}}) \|_2^2 + \| \hat{\boldsymbol{u}}_k^{p_i} - \boldsymbol{\pi}(\boldsymbol{T}_b^C\boldsymbol{T}_{b_i}^{b_{i-1}}\boldsymbol{T}_C^b\boldsymbol{p}_k^{C_i}) \|_2^2 \end{aligned} \tag{12}$$

In the equation, $\hat{u}_k^{p_{i-1}}$ represents the observed coordinates of a feature point in the pixel coordinate system of the image frame $F_i$, and $u_k^{p_{i-1}}$ represents the theoretical coordinates of the feature point $k$ n the pixel coordinate system of the image frame $F_i$, which corresponds to the inter-frame projection result.

In computer vision and 3D reconstruction, it is commonly assumed that the reprojection error follows a Gaussian distribution [27,28]. Based on this assumption, the probability of a feature point's movement can be calculated using the reprojection error. Assuming the Gaussian distribution has a mean of $\mu_0$, the movement probability of the feature point can be described as Equation (13):

$$p = \frac{1}{\sqrt{2\pi}\sigma_0} \int_{-\frac{r}{2}}^{\frac{r}{2}} e^{-\frac{(x-\mu_0)^2}{2\sigma_0^2}} dx \tag{13}$$

In the equation, $r$ represents the bidirectional reprojection error of the feature point. It can be observed that, as the reprojection error increases, the probability of the feature point's movement also increases. To avoid the influence of IMU preintegration errors on the reprojection error, this paper proposes a dynamic adaptive method for determining the threshold, which includes methods for determining the mean and standard deviation. In theory, the reprojection error of a completely static feature point should be zero. However, due to noise in IMU measurements and distortion in the camera, the reprojection error is not zero. Therefore, this paper designs a dynamic adaptive loss function to determine the mean and standard deviation of the Gaussian distribution; the calculation method is given by Equation (14):

$$l(s(f), \hat{s}(f)) = -(s(f)\log(\hat{s}(f)) + (1 - s(f))\log(1 - \hat{s}(f))) \tag{14}$$

In the equation, $s(f) = \{0, 1\}$ represents the label value, where 0 indicates a static feature point and 1 indicates a dynamic feature point, and $\hat{s}(f)$ represents the predicted probability of the motion state of the feature point, denoted as $s(f) = p(r_f)$.

Assuming the extracted set of absolutely static points in the image $I$ is denoted by $\Omega_{static} = \{f_1, f_2, \ldots, f_m\}$, the set of features with absolute motion is denoted by $\Omega_{Dynamic} = \{f_{m+1}, f_{m+2}, \ldots, f_{m+n}\}$, and the set of temporarily static features is denoted by $\Omega_{temporary} = \{f_{m+n+1}, f_{m+n+2}, \ldots, f_N\}$. Since both the absolutely static and absolutely moving features can obtain accurate probability values using semantic labels, for each frame of the image, the semantic information can be used to determine the estimated samples, and the expression is given by Equation (15):

$$s(f) = \begin{cases} 0, f \in \Omega_{static} \\ 1, f \in \Omega_{Dynamic} \end{cases} \tag{15}$$

After computing the reprojection error for each feature point, it is necessary to first utilize the semantic information to provide estimated samples for the aforementioned binary classification problem. Subsequently, using the sample information, the objective is to minimize the dynamic adaptive loss function in order to obtain the estimated mean and standard deviation values. The optimization objective can be described by Equation (16).

$$\min_{\mu,\sigma}\left\{-\left(s(f)\log\left(\frac{1}{\sqrt{2\pi}\sigma}\int_{-\frac{r_f}{2}}^{\frac{r_f}{2}} e^{-\frac{(x-\mu)^2}{2\sigma^2}} dx\right) + (1-s(f))\log\left(1 - \frac{1}{\sqrt{2\pi}\sigma}\int_{-\frac{r_f}{2}}^{\frac{r_f}{2}} e^{-\frac{(x-\mu)^2}{2\sigma^2}} dx\right)\right)\right\} \tag{16}$$

By solving the above optimization problem, we can obtain the mean and standard deviation that minimize the cross-entropy loss [29]. Combining Equation (13), we can obtain the probability of movement for temporary stationary feature points. In some special cases, such as when the carrier motion environment is stationary or there are no feature points in absolute motion, the prior semantic information can only provide single-class sample information. In such cases, it is not possible to obtain a reasonable mean and standard

deviation by minimizing the loss function. In this case, it is only necessary to estimate the mean and standard deviation using the absolute static features in the environment. The reprojection error of absolute static features in the environment can be used as samples, and the maximum likelihood estimation method can be used to estimate the mean and variance of the Gaussian distribution. Then, using Equation (13), the movement probability of other feature points can be calculated. The algorithm framework is shown in Algorithm 1.

---

**Algorithm 1** Motion consistency detection algorithm based on visual–inertial coupling

---

**Input:** current frame $F_i$, previous adjacent frame $F_{i-1}$, set of absolute static points in the current frame $\mathbf{\Omega}_{static}$, set of absolute motion feature points in the current frame $\mathbf{\Omega}_{Dynamic}$, set of temporarily static feature points in the current frame $\mathbf{\Omega}_{temporary}$, IMU pre-integration results between consecutive frames $I_{i-1,i}$.

**Output:** probability of feature point set movement in the current frame $P = \{p_{f_1}, p_{f_2}, \ldots, p_{f_N}\};$

1: **for** $f \in \mathbf{\Omega}_{static} \bigcup \mathbf{\Omega}_{temporary} \bigcup \mathbf{\Omega}_{Dynamic}$ **do**
2:     $r(f) = Calculatereprojection(f_{i-1}, f_i, I_{i-1,i});$
3: **end for**
4: **if** $\mathbf{\Omega}_{Dynamic} = \oslash$ **then**
5:     $(\mu, \sigma) = MLE(\mathbf{\Omega}_{static}, r_{f_{static}});$
6:     **for** $f_i \in \mathbf{\Omega}_{temporary}$ **do**
7:         $p_{f_i} = \frac{1}{\sqrt{2\pi}\sigma} \int_{-\frac{r_{f_i}}{2}}^{\frac{r_{f_i}}{2}} e^{-\frac{(x-\mu)^2}{2\sigma^2}} dx$
8:     **end for**
9: **else**
10:     **for** $f_i \in \mathbf{\Omega}_{static} \bigcup \mathbf{\Omega}_{Dynamic}$ **do**
11:         $dict.insert(\{s(f_i), p_{f_i}(r_{f_i}, \mu, \sigma)\});$
12:     **end for**
13:     $(\bar{\mu}, \bar{\sigma}) = \arg \min(loss(s(f), \hat{s}(f)), dict)$
14:     **for** $f \in \mathbf{\Omega}_{temporary}$ **do**
15:         $p_{f_i} = \frac{1}{\sqrt{2\pi}\bar{\sigma}} \int_{-\frac{r_{f_i}}{2}}^{\frac{r_{f_i}}{2}} e^{-\frac{(x-\bar{\mu})^2}{2\bar{\sigma}^2}} dx$
16:     **end for**
17: **end if**

---

Figures 5 and 6 compare the results of the coarse segmentation algorithm with the feature-extraction results after IMU correction.

In test image 1, there are three individuals, with only one customer in motion, while the other two individuals are in a stationary position with minimal movement. However, in the motion coarse segmentation algorithm based on deep learning, the prior probability of motion assigned to the "person" feature attribute is high, resulting in misclassification of feature points falling on stationary individuals as dynamic points (see Figure 5a). In contrast, due to the introduced IMU-assisted correction method in the motion-consistency-detection algorithm, some of the misclassifications caused by coarse segmentation are corrected. As shown in Figure 5b, the number of dynamic features on the stationary customer is significantly reduced. As shown in Figure 6a, the YOLOv8 network detects four human objects, but the leftmost individual is not a real human, but a billboard, and the rightmost individual is just a mannequin. However, in the motion-coarse-segmentation algorithm, the feature points falling on these two objects are labeled as dynamic features. In comparison, the proposed algorithm in this section has fewer misclassifications. The number of dynamic features mentioned on the billboard and mannequin is significantly reduced, almost zero. As shown in Figure 6b, two customers in the middle, who did not undergo significant movement, should also have a corresponding reduction in the number of dynamic features on them.

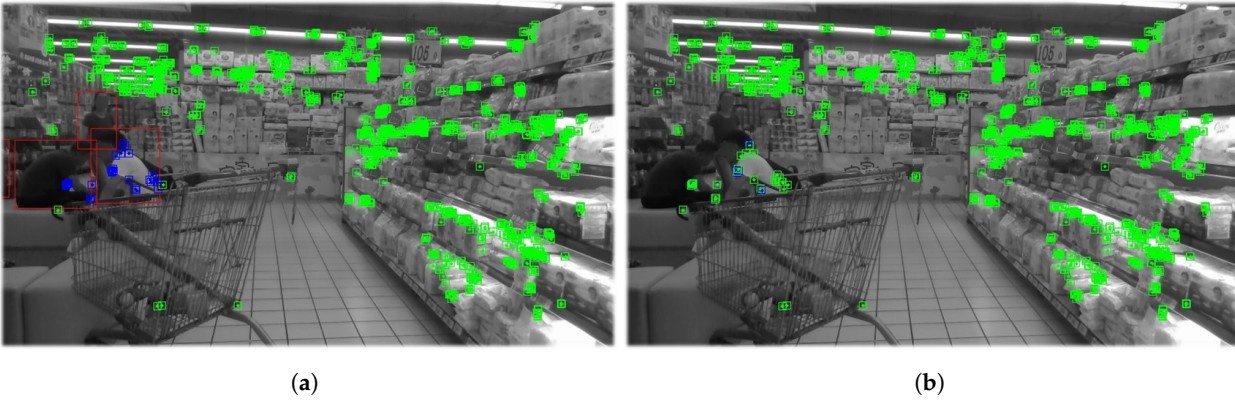

(**a**)                    (**b**)

**Figure 5.** Shows the comparison of feature selection results for different algorithms on test image 1. (**a**) The coarse segmentation algorithm; (**b**) the corrected results.

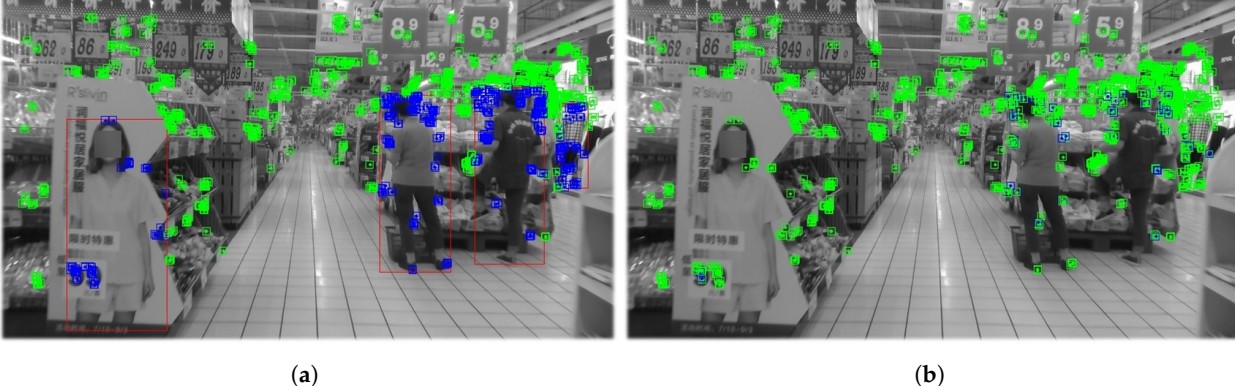

(**a**)                    (**b**)

**Figure 6.** Shows the comparison of feature selection results for different algorithms on test image 2. (**a**) The coarse segmentation algorithm; (**b**) the corrected results.

## 4. Results

In the previous sections, the fundamental theory of the VSLAM localization algorithm proposed in this paper was introduced. In this section, experiments will be designed to validate the performance of the proposed algorithm. The experiments are divided into three main parts. Firstly, the performance of the deep learning-based coarse segmentation algorithm will be validated using the TUM dataset. Since this dataset does not include IMU measurements, the OpenLORIS [30] dataset will be used to validate the performance of the feature optimization algorithm based on visual–inertial coupling. Finally, the feasibility of the proposed algorithm will be verified through real-world environment experiments.

### 4.1. TUM Dataset Testing Experiment

The TUM RGB-D dataset provides several image sequences containing dynamic objects in indoor environments. In this paper, we selected the high dynamic range dataset, specifically the fr3/walking sequence, to evaluate the performance of our coarse segmentation algorithm. The semantic information extraction algorithm we used is based on the YOLOv8 network for object detection. The experiments were conducted on a Lenovo laptop, which has an Intel Core i7 CPU processor with a clock speed of 5.0 GHz. It has 14 cores and 20 threads, along with 16 GB of DDR5 memory operating at a frequency of 5200 MHz. The GPU is an NVIDIA GeForce RTX 4060 with 8 GB of VRAM. With TensorRT acceleration on this device, the semantic information extraction time per frame is approximately 33.3 ms.

The algorithm proposed in this paper is an improvement over the ORB-SLAM3 algorithm [31]. To validate the performance of the improved algorithm, we compared it with the ORB-SLAM3 algorithm in terms of localization accuracy on the dataset. The results are shown in Figure 7. Additionally, we also calculated the root-mean-squared error (RMSE)

for both algorithms in terms of the absolute trajectory error (APE) and relative trajectory error (RPE) [32], and the results are summarized in Tables 1–3.

From the above table, it can be observed that the algorithm proposed in this paper demonstrates improvements over the ORB-SLAM3 algorithm in all four performance evaluation metrics, including the ATE, RPE rotation, and RPE translation. In terms of the ATE, the RMSE improvement rates for the four image sequences are 88.5%, 78.93%, 61.54%, and 95.31%, respectively, all exceeding 60%. This indicates a significant enhancement in performance. Therefore, we can conclude that dynamic objects significantly degrade the localization accuracy of ORB-SLAM3, while the proposed algorithm effectively identifies and removes dynamic objects in the environment, thereby improving the pose estimation accuracy.

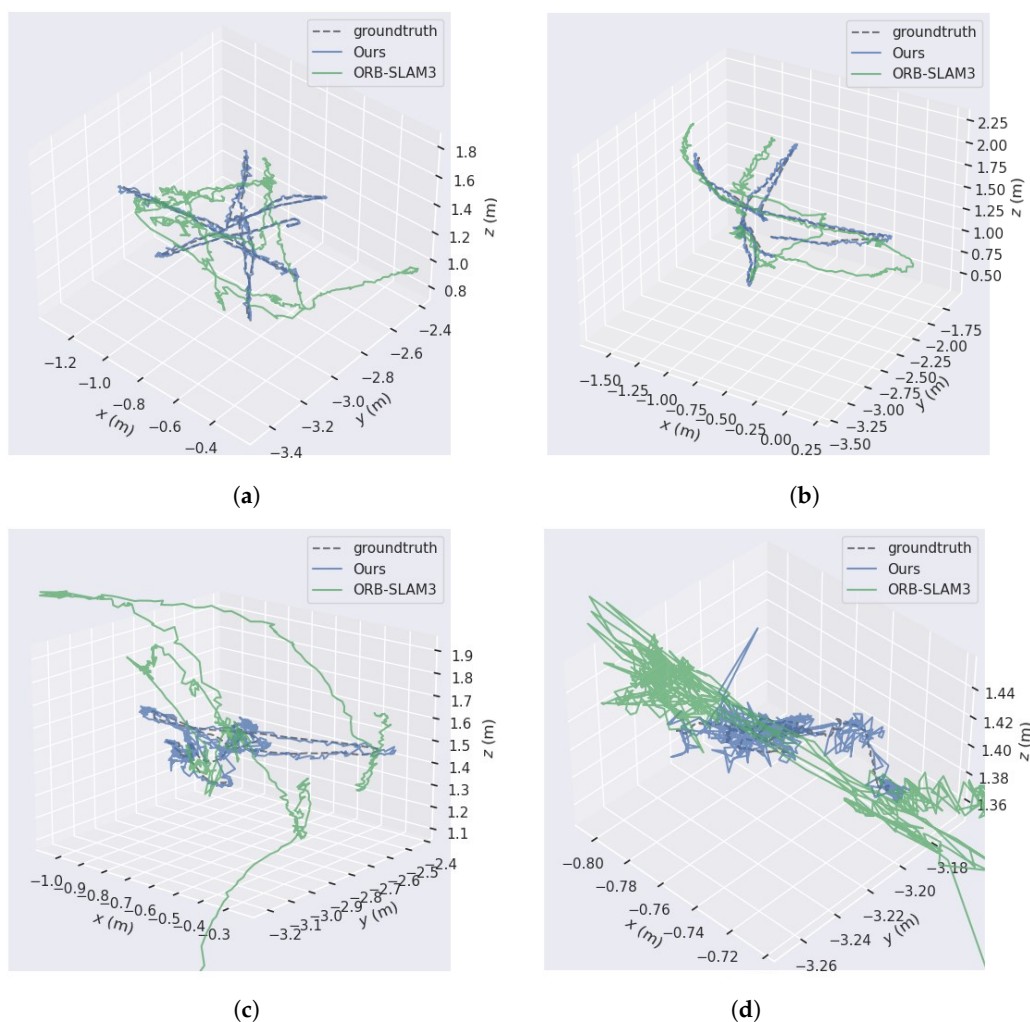

**Figure 7.** Results of the (**a**) fr3/walking/xyz, (**b**) fr3/walking/halfsphere, (**c**) fr3/walking/rpy, and (**d**) fr3/walking/static sequences for the TUM dynamic RGB-D datasets. We compared the trajectories estimated by our methods and by the original ORB-SLAM3 with the ground truth.

**Table 1.** ATE (in meters) for our method and the original ORB-SLAM3.

| Algorithms | Sequences | | | |
| --- | --- | --- | --- | --- |
| | fr3/w/halfsphere | fr3/w/rpy | fr3/w/static | fr3/w/xyz |
| ORB-SLAM3 | 0.1831 | 0.1552 | 0.0229 | 0.2793 |
| Ours | 0.0206 | 0.0327 | 0.0088 | 0.0131 |
| Improvement (100%) | 88.75 | 78.93 | 61.57 | 95.31 |

**Table 2.** Translational drift of RPE (in meters) for our method and the original ORB-SLAM3.

| Algorithms | Sequences | | | |
| --- | --- | --- | --- | --- |
| | fr3/w/halfsphere | fr3/w/rpy | fr3/w/static | fr3/w/xyz |
| ORB-SLAM3 | 0.0147 | 0.0199 | 0.0062 | 0.0118 |
| Ours | 0.0130 | 0.0056 | 0.0023 | 0.0095 |
| Improvement (100%) | 11.56 | 71.85 | 62.90 | 19.49 |

**Table 3.** Rotational drift of RPE (in meters) for our method and the original ORB-SLAM3

| Algorithms | Sequences | | | |
| --- | --- | --- | --- | --- |
| | fr3/w/halfsphere | fr3/w/rpy | fr3/w/static | fr3/w/xyz |
| ORB-SLAM3 | 0.0126 | 0.0169 | 0.0091 | 0.0140 |
| Ours | 0.0105 | 0.0134 | 0.0045 | 0.0097 |
| Improvement (100%) | 16.67 | 20.71 | 50.55 | 30.71 |

*4.2. OpenLORIS Dataset Experiment*

To validate the performance of the proposed visual–inertial tightly coupled motion segmentation algorithm, in this section, the OpenLORIS-Scene-market sequence is utilized to assess the localization accuracy of our algorithm. This sequence was captured using an Intel Realsense D400 camera in a supermarket environment. The scene contains a significant number of pedestrians, including some who are stationary or have minimal movement. This provides an opportunity to evaluate the localization precision of our algorithm. Due to the disparate sampling rates of the accelerometer and gyroscope data in this dataset, linear interpolation is employed to align the accelerometer and gyroscope data to a consistent frequency for convenient testing of our algorithm in this study.

To validate the feasibility of the algorithm, this section conducts experiments based on the market dataset and compares the localization results of three algorithms: ORB-SLAM3, Coarse Segment SLAM (CS-SLAM) based on deep learning for motion coarse segmentation, and our method (VIS-SLAM) based on visual–inertial tightly coupled feature optimization. Figure 8 illustrates the localization results of the three algorithms on the market dataset.

To facilitate a clear comparison of the localization accuracy of the three algorithms, Figure 9 illustrates the absolute errors in the XYZ directions (in the local world coordinate system) for each algorithm. In order to provide a qualitative description of the errors, Table 4 presents the root-mean-squared errors (RMSEs) for each algorithm in different directions.

From the experimental results, it can be observed that, compared to ORB-SLAM3 and the motion coarse segmentation algorithm based on YOLOv8, the proposed algorithm demonstrates improved localization accuracy in all three directions. This is because the presence of numerous pedestrians in the supermarket scene breaks the assumption of a static environment, leading to a decrease in the accuracy of the ORB-SLAM3 algorithm. However, the motion coarse segmentation algorithm may mistakenly classify some static features as dynamic features and remove them, resulting in a decrease in the number of features and, subsequently, reducing the localization accuracy. In contrast, the proposed algorithm effectively corrects the misclassifications introduced by the coarse segmentation algorithm, thereby improving the localization accuracy.

**Table 4.** Root mean squared errors (RMSEs) of different algorithms in different directions.

| Algorithms | Direction | | |
|---|---|---|---|
| | *x* | *y* | *z* |
| ORB-SLAM3 | 1.6713 | 1.2933 | 2.8324 |
| CS-SLAM | 1.1088 | 0.7695 | 1.2876 |
| VIS-SLAM | 0.7768 | 0.7616 | 0.9048 |

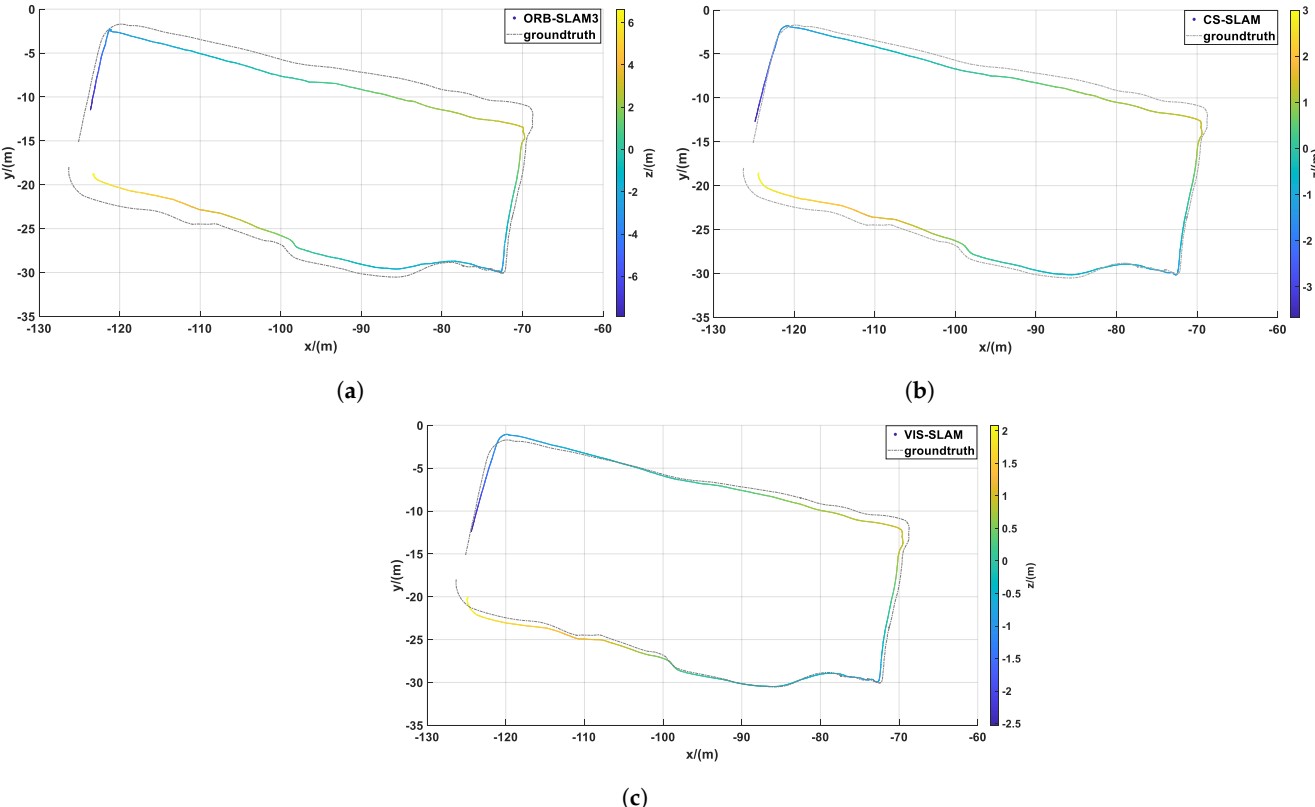

**Figure 8.** Comparison of localization trajectory results for (**a**) ORB-SLAM3, (**b**) CS-SLAM, and (**c**) VIS-SLAM.

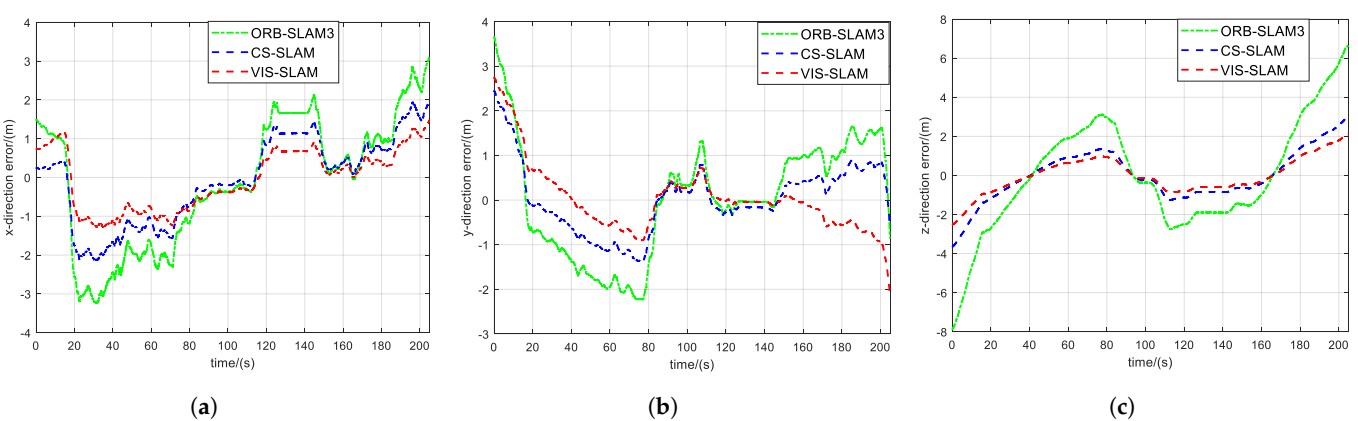

**Figure 9.** Error comparison of different algorithms in different directions. (**a**) x-direction; (**b**) y-direction; (**c**) z-direction.

## 4.3. Real-World Experiment

To ensure a dynamic experimental environment, a campus with high pedestrian traffic was selected as the experimental scene for this study. The mobile experimental setup traversed areas of high foot traffic, including student dormitories, cafeterias, supermarkets, and laboratory buildings. This choice of locations allows for a better validation of the proposed algorithm in this paper. Figure 10 shows some scenes from the experiments.

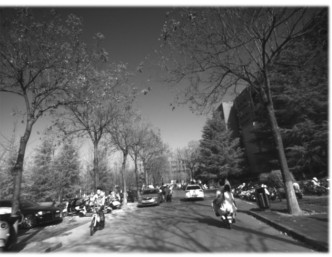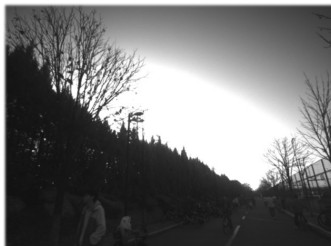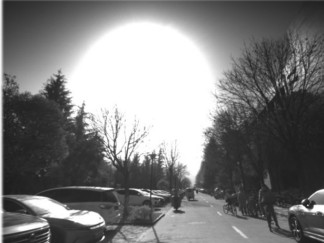

**Figure 10.** Partial scenes from the experimental environment.

As shown in Figure 11a,b, the hardware devices and deployment framework for the experimental setup are illustrated. In this experiment, all sensor data were communicated and shared through the Robot Operating System (ROS) [33]. The image data were acquired using a camera sensor, specifically the Realsense D435i camera, in monocular mode. The left-eye image information was utilized, with an image resolution of 640 × 480 and an input frequency of 30 Hz. The IMU information was provided by the built-in IMU module of the Realsense D435i camera. The output frequencies of the accelerometer and gyroscope were both set to 200 Hz. The u-blox module, in combination with the NTRIP module, provided corrected latitude, longitude, and altitude information.

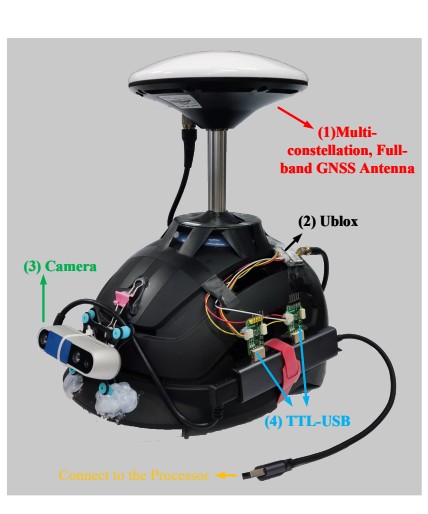

(**a**)

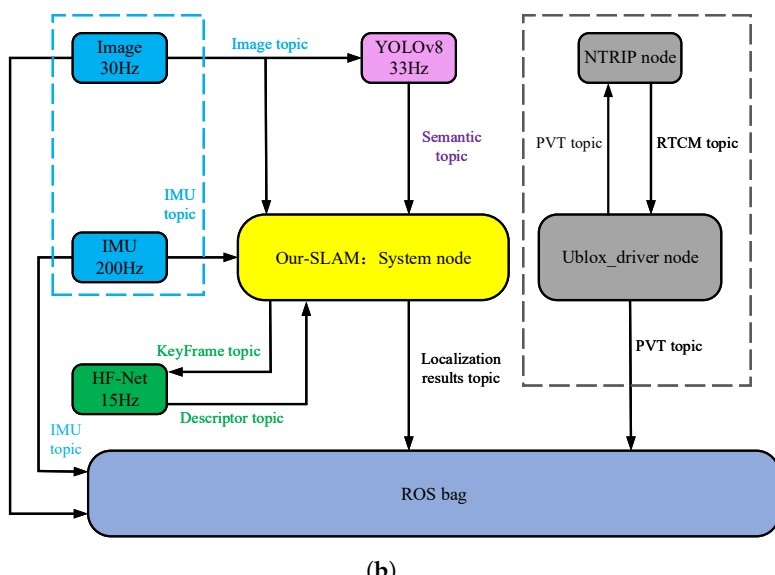

(**b**)

**Figure 11.** Diagram of experimental hardware platform and software framework. (**a**) Hardware device; (**b**) software framework.

In this paper, real-time RTK data were used as the ground truth for comparing the localization accuracy between ORB-SLAM3 and the proposed algorithm. Figures 12 and 13 present the localization results of the two algorithms and the error comparison in the three-axis directions. Table 5 presents the root-mean-squared errors (RMSEs) for each algorithm in different directions.

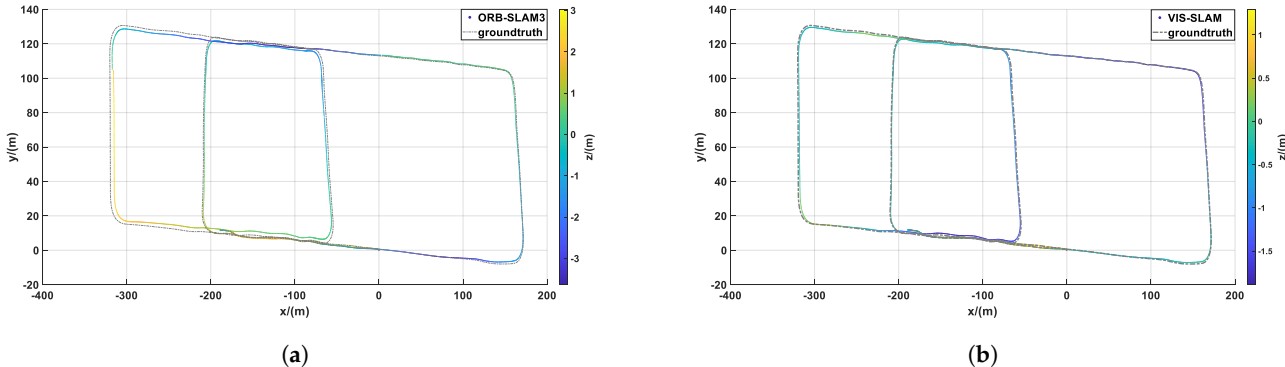

(**a**)　　　　　　　　　　　　　　　　　　　　(**b**)

**Figure 12.** Comparison of localization results between the two algorithms. (**a**) ORB-SLAM3; (**b**) VIS-SLAM.

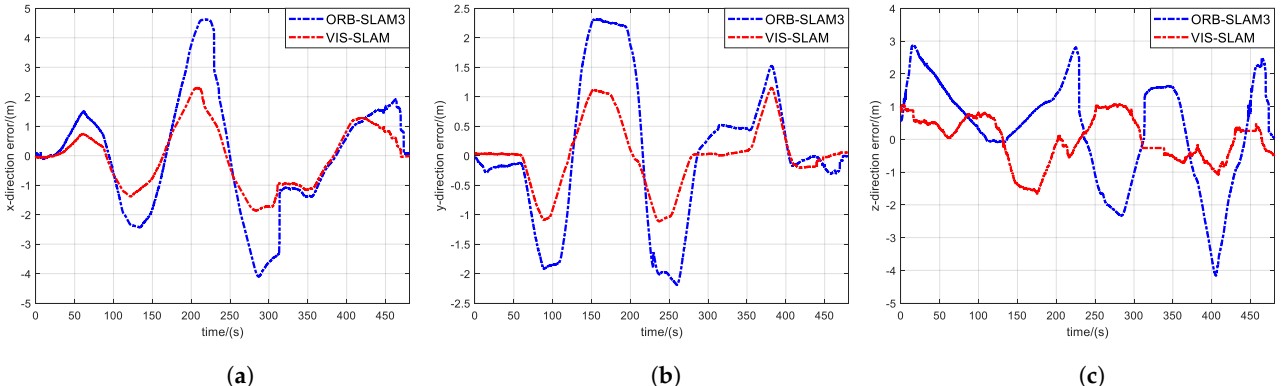

(**a**)　　　　　　　　　　　　(**b**)　　　　　　　　　　　　(**c**)

**Figure 13.** Error comparison of different algorithms in different directions. (**a**) x-direction; (**b**) y-direction; (**c**) z-direction.

**Table 5.** Root mean squared errors (RMSEs) of different algorithms in different directions.

| Algorithms | Direction | | |
|---|---|---|---|
| | *x* | *y* | *z* |
| ORB-SLAM3 | 2.0080 | 1.2287 | 1.6232 |
| VIS-SLAM | 1.0456 | 0.5610 | 0.7076 |
| Improvement (100%) | 47.93 | 54.34 | 56.40 |

From the experimental results, it can be observed that the proposed algorithm exhibits significant improvements in localization accuracy compared to the ORB-SLAM3 algorithm. This indicates that the proposed algorithm effectively suppresses the interference of dynamic objects in the environment. The errors in all three directions for the proposed algorithm are less than 1.0 m, with accuracy improvement rates of 47.93%, 54.34%, and 56.40% compared to the ORB-SLAM3 algorithm in the respective directions. Comparing the 3D localization trajectories of the two algorithms, it can be seen that, in certain sections, the ORB-SLAM3 algorithm exhibits larger drift. This is due to the presence of numerous pedestrians and vehicles in these sections, which disrupts the assumption of a static environment and leads to a decrease in localization accuracy. However, the proposed algorithm effectively removes dynamic features, resulting in significantly improved accuracy and reduced drift in such sections.

To validate the operational efficiency of the non-blocking model proposed in the paper, a comparison was made between the processing time per frame of image in the semantic information-extraction thread for both the blocking and non-blocking models, under the condition that the hardware devices and the semantic information extraction network are

identical. The unit of measurement is milliseconds (ms), and the comparison results are presented in Table 6.

**Table 6.** Comparison of running times of different algorithms in various stages.

|  | Feature | Semantic | Average Latency |
|---|---|---|---|
| ORB-SLAM3 | $12.21 \pm 4.65$ | / | / |
| blocking Model | $12.21 \pm 4.65$ | $33.36 \pm 2.18$ | 21.15 |
| Non-blocking Model | $12.21 \pm 4.65$ | $33.36 \pm 2.18$ | 7.74 |

From the experimental results, it can be observed that the feature extraction time is consistent across the original ORB-SLAM3 algorithm, the version with the blocking model added, and the version with the non-blocking model added. For both models, since the YOLOv8 network is used to extract semantic information, the semantic information extraction time per frame is the same. However, under the blocking model, each frame of the image has to wait for the semantic result, resulting in an average extension time of 21.15 ms. In contrast, under the non-blocking model, only the latest frame in the sliding window needs to wait for the semantic information after feature extraction, which means approximately one frame out of every three needs to wait. Therefore, the average extension time is shorter, only requiring 7.74 ms. It can be seen that the non-blocking model can greatly improve the efficiency of pose estimation calculation.

## 5. Conclusions

This paper proposes a real-time dynamic SLAM technique based on deep learning and visual–inertial sensing. To improve system real-time performance, a non-blocking semantic information extraction method is designed. Under this framework, YOLOv8 object detection is utilized to extract semantic information from image frames. Based on a hierarchical probabilistic motion model, feature points are assigned different prior motion probabilities. Additionally, a motion probability propagation model is employed to compensate for the inability to extract semantic information from all image frames. To avoid misclassifying static features as dynamic features and removing them, a feature-optimization algorithm based on visual–inertial coupling is introduced. By calculating inter-frame pose transformations using IMU preintegration results, the reprojection errors of feature points are obtained. Furthermore, an adaptive threshold-based feature-selection algorithm is designed to further enhance the localization accuracy. The experimental results demonstrate that the proposed algorithm achieves good localization accuracy in dynamic environments while ensuring real-time performance of the system.

In future work, due to the coarse semantic information extracted by the object-detection network from images, we hope to introduce a lightweight semantic segmentation network to improve the accuracy of the prior probabilities of feature points. At the same time, the motion-probability-grading model proposed in this paper is given based on life experience, and it may not be applicable in some special scenarios. Therefore, we hope to propose a visual–semantic–inertial tightly coupled feature optimization method in future work, which will perform bidirectional screening of features to enhance the robustness of the algorithm.

**Author Contributions:** Methodology, Minkun Zhao; formal analysis, Yinglong Wang; investigation, Xinlong Xu; writing—original draft preparation, Yinglong Wang; writing—review and editing, Yinglong Wang; supervision, Xiaoxiong Liu. All authors have read and agreed to the published version of the manuscript.

**Funding:** This research was funded by the National Natural Science Foundation of China, grant number 61573286, and the Aeronautical Science Foundation of China, grant number 201905053003.

**Informed Consent Statement:** Informed consent was obtained from all subjects involved in the study.

**Data Availability Statement:** Publicly available datasets were analyzed in this study. The TUM dataset can be found at http://vision.in.tum.de/data/datasets/rgbd-dataset, and the OpenLORIS dataset can be found at https://lifelong-robotic-vision.github.io/dataset/scene.

**Acknowledgments:** The authors would like to thank the Editor, Associate Editor, and anonymous reviewers for processing our manuscript.

**Conflicts of Interest:** The authors declare no conflicts of interest.

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
