# Peer review of "VIS-SLAM: A Real-Time Dynamic SLAM Algorithm Based on the Fusion of Visual, Inertial, and Semantic Information"

_ijgi, doi:10.3390/ijgi13050163_

Round 1
Reviewer 1 Report
Comments and Suggestions for Authors
The paper proposes a dynamic visual-inertial SLAM that integrates semantic segmentation. The authors design a non-blocking model for extracting semantic information from images, utilizing a motion probability propagation model to assign prior motion probabilities to features. They further devise an adaptive threshold-based feature selection algorithm to eliminate dynamic objects, thus enhancing the localization accuracy of the visual-inertial system. Ultimately, the superiority of the proposed method is substantiated through both datasets and real-world experiments. The practical value in visual-inertial SLAM makes this paper potentially acceptable with minor modifications. Here are some suggestions:
1. Given that the Abstract mentions that a non-blocking model for semantic information extraction from images was devised due to real-time constraints imposed by deep learning algorithms, it is suggested that a comparison of algorithmic real-time performance should be included in the experiments.
2. In Figure 3, the motion probability propagation model for dynamic targets, it is recommended to clarify the specific classification criteria adopted.
3. The annotation of the proposed algorithm in Table 4 differs from those in Tables 1, 2, and 3; therefore, it is advised to harmonize the description across all tables.
4. For Figure 11, it is suggested to add labels and descriptions for each of the sensors in the hardware setup to enhance clarity.
Comments on the Quality of English LanguageNone
Author Response
Dear Editors and Reviewers:
Thank you for your letter and for the reviewers’ comments concerning our manuscript entitled “VIS-SLAM: A Real-Time Dynamic SLAM Algorithm Based on Fusion of Visual, Inertial, and Semantic Information” (ID: ijgi-2952824).Those comments are all valuable and very helpful for revising and improving our paper, as well as the important guiding significance to our researches. We have studied comments carefully and have made correction which we hope meet with approval.

Reviewer 2 Report
Comments and Suggestions for Authors
The paper covers the important topic of simultaneous localization and mapping (SLAM). It proposes a Visual Inertial SLAM technique and integrates motion probability estimates to the feature points. The algorithm has been tested on two publicly available datasets and has been compared to the well-known ORB-SLAM3 algorithm. Significant performance increases are shown.
In general, the paper is well written, necessary formulas are given. The concept of the algorithm is clearly described. Diagrams as well as tables are used to show the performance. However, let me make a few comments and proposals. For a better understanding it will be helpful, if the authors explicitly describe how the performance of the algorithm will decrease in case that there is a larger number of objects in the scene , which cannot be classified, since they were not learned, or which are misclassified. Since this will result in wrong static/dynamic object classifications, performance will decrease. How will it then compare to ORB-SLAM or other algorithms?
The framework diagram does not clearly show, how the new approach is integrated.
It will be helpful, it error measures are explicitly defined.
The authors compare their approach to ORB-SLAM3 on two test databases and to CS-SLAM in a second test. It is not clear, why these algorithms were chosen for comparison? Are they best in class? As the authors mention themselves in teh introduction, there are various SLAM algorithms available. Thus, the authors show, that there is a significant improvement compared to these algorithms. But it is not clear, how well the new approach performs compared to best in class algorithms.
The algorithm has been tested on two different databases. Unfortunately, the presentation (diagrams, tables) of the results changes from one test to the other. I recommend to find a format that fits both tests.
Comments on the Quality of English Language
Quality of the English language is good. There are a few typing errors and minor grammar errors in the text.
Author Response

(The authors gave the same response as above.)

Reviewer 3 Report
Comments and Suggestions for Authors
This paper overall presents a new approach to improving the robustness and accuracy of visual-inertial simultaneous localization and mapping (SLAM) systems in dynamic environments. The authors have identified some key limitations of existing SLAM methods and have proposed reasonable solutions to address these issues. The combination of deep learning-based semantic segmentation by using YOLO, visual-inertial integration, and adaptive motion consistency detection is a promising approach for distinguishing and handling dynamic objects in the environment.
Strengths:
- The non-blocking model for semantic information extraction is a clever design that aims to improve the real-time performance of the system by decoupling the feature extraction and semantic segmentation processes.
- The hierarchical motion probability model and the motion probability propagation model provide a principled way to leverage semantic information for coarse motion segmentation.
- The use of visual-inertial tight coupling and adaptive thresholding for motion consistency detection is a well-justified approach to refine the coarse segmentation results and improve localization accuracy.
- The experimental evaluation on publicly available datasets (TUM RGB-D and OpenLORIS) as well as real-world experiments demonstrates the effectiveness of the proposed approach in terms of localization accuracy and robustness to dynamic objects.
Suggestions:
- The paper would benefit from a more comprehensive literature review and a clearer positioning of the proposed work in relation to existing dynamic SLAM methods, especially those that also leverage visual-inertial fusion and deep learning.
- The description of the motion probability grading model should be more detailed, with a clearer justification for the chosen categories and probability values.
- The experimental section should be expanded to include more quantitative comparisons with state-of-the-art dynamic SLAM methods, both in terms of localization accuracy and computational performance.
- Discussion of potential limitations or failure cases would provide a more balanced perspective.
In general, this paper presents a solid contribution to the field of dynamic SLAM, with a new approach that combines deep learning, visual-inertial fusion, and adaptive motion consistency detection. With some additional clarifications, expanded experiments, and a more comprehensive literature review, this work could be a valuable addition to the existing body of research in this area.
Comments on the Quality of English LanguageModerate editing of English language required
Author Response

(The authors gave the same response as above.)

Round 2
Reviewer 3 Report
Comments and Suggestions for Authors
I checked their updates on the revised manuscript, and the authors resolved my previous comments in the revised manuscript.
Comments on the Quality of English LanguageMinor editing of English language required